# Experimental Investigation of Concrete Sandwich Walls with Glass-Fiber-Composite Connectors Exposed to Fire and Mechanical Loading

Marcin Haffke, Matthias Pahn *, Catherina Thiele and Szymon Grzesiak

Department of Civil Engineering, Technical University of Kaiserslautern, 67663 Kaiserslautern, Germany; marcin.haffke@bauing.uni-kl.de (M.H.); catherina.thiele@bauing.uni-kl.de (C.T.); szymon.grzesiak@bauing.uni-kl.de (S.G.)
* Correspondence: matthias.pahn@bauing.uni-kl.de

**Abstract:** Precast concrete sandwich panels (PCSPs) are known for their good thermal, acoustic and structural properties. Severe environmental demands can be met by PCSPs due to their use of highly thermally insulating materials and non-metallic connectors. One of the main issues limiting the wider use of sandwich walls in construction is their unknown fire resistance. Furthermore, the actual behaviour of connectors and insulation in fire in terms of their mechanical performance and their impact on fire spread and the fire resistance of walls is not fully understood. This paper presents an experimental investigation on the structural and thermal behaviour of PCSPs with mineral-wool insulation and glass-fiber-reinforced polymeric bar connectors coupling two concrete wythes. Three full-size walls were tested following the REI certification test procedure for fire walls under fire and vertical eccentric and post-fire mechanical impact load. The three test configurations were adopted for the assessment of the connectors' fire behaviour and its impact on the general fire resistance of the walls. All the specimens met the REI 120-M criteria. The connectors did not contribute to the fire's spread and the integrity of the walls was maintained throughout the testing time. This was also confirmed in the most unfavourable test configuration, in which some of the connectors in the inner area of the wall were significantly damaged, and yet the structural connection of the concrete wythes was maintained. The walls experienced heavy heat-induced thermal bowing. The significant contribution of connectors to the stiffness of the wall during fire was observed and discussed.

**Keywords:** fibre-reinforced polymer connectors; concrete sandwich panels; fire test; fire resistance; fire wall; concrete composite structures

## 1. Introduction

Pre-cast concrete sandwich panels (PCSP) are known for their good inherent thermal and acoustic insulating properties. This technology allows the fast construction of energy-efficient and durable buildings. Due to these advantages and the growing need for sustainable solutions, sandwich panels have gained in popularity in industrial and residential construction. Pre-cast concrete sandwich panels are usually composed of two steel-reinforced concrete wythes and an insulation layer in between. The outer concrete layer (facing wythe) usually does not play any structural role and serves solely as a form of aesthetic protection for the insulating layer [1]. The inner, load-bearing concrete wythe is designed to carry all the vertical and bending loads. The two concrete wythes are mechanically linked together by means of connectors. The connectors and insulation together constitute the so-called core layer [2]. There is a wide range of connector types available on the market [2,3]. The types of connector that are increasingly applied in modern sandwich panels are various products of fibre-reinforced polymer (FRP) composite. They show good mechanical properties together with excellent durability [4,5] and thermal characteristics [6].

Thus, due to the reduction in thermal bridging, they increase the thermal characteristics of the entire wall.

In recent years, different sandwich-panel elements were tested on their mechanical behaviour. For example, the studies of Schultz-Cornelius [7,8] and Pahn [3] contributed to the understanding of the behaviour of PCSP elements. Lameiras et. al. [9] showed the experimental results of four almost-full-scale sandwich panels under in-plane cyclic loading. This investigation contributed to the literature about the seismic behaviour of precast concrete elements. Moreover, the study focused on critical regions with FRP connectors in the form of rectangular plates with thicknesses of 2.0 mm. Lezgy-Nazargah et. al. [10] presented a simple 1D nonlinear finite element model for the analysis of concrete sandwich-panel structures. The reason for undertaking these studies is the abundance of 2D and 3D models and the necessity of filling this gap with a new displacement-based, materially non-linear 1D finite element model with a low number of degrees of freedom.

The wide use of PCSPs in construction, as well as for applications requiring fire resistance, prompts concerns regarding the fire behaviour of their components. This hinders the potential use of sandwich walls as fire walls. Estimating their fire resistance is mostly limited by a lack of knowledge regarding the fire behaviour of non-metallic connectors. The degradation of their stiffness and strength, as well as their anchorage capacity and the potential damage to concrete wythes by melted or pulled-out connectors, constitute dangers to the fire resistance of the entire sandwich wall. Due to a huge number of products (e.g., various forms and materials used), reliable guidelines for the estimation of the fire behaviour of glass-fibre-reinforced polymeric (GFRP) connectors are not easy to develop. Another difficulty is the applicability of the standard assessment methods described in testing codes that are tailored to flat-surface materials [11]. Therefore, it would be favourable to test the fire performance of non-metallic connectors in conditions similar to those present in a realistic application of a sandwich wall, including limited oxygen influx and under load. Considering the above-mentioned limitations and needs, laboratory fire tests deliver appropriate conditions for the realistic description of the fire behaviour of GFRP connectors and, thus, of whole sandwich walls in the presence of fire.

Even though the insulation layer and facing wythe are not usually considered as load-bearing layers in a structural design, the mechanical coupling of both concrete wythes, which is caused by the transfer of shear forces between them through connectors and insulation, can significantly contribute to the overall stiffness and strength of the panel [12–15]. The degree of composite action (DCA) describes the level of mechanical coupling of concrete wythes. The higher the DCA, the stiffer the sandwich panel and, in terms of stress distribution, its structural behaviour resembles that of a monolithic member [16]. High DCA is favourable in terms of the flexural stiffness and strength of panels and enhances their stability. As mentioned in work on the fire performance of steel sandwich panels [17], due to their various constituents, sandwich-panel walls exhibit different behaviour in the presence of fire to their pure components and should therefore be assessed as a whole system regarding their flammability. However, according to the authors' knowledge, no design codes and very limited work [13,15] addressing the behaviour of sandwich panels and their degree of composite action under fire load can be found. Since high DCA activates the facing wythe and insulation layer in flexure, causing them to be more stressed, its influence on structural performance at elevated temperatures should be investigated in depth. Therefore, the accurate assessment of the contribution of the temperature-dependent core layer's shear stiffness would lead to the less conservative and more efficient design of PCSPs. The consideration of this contribution to the design of PCSPs could increase their predicted fire resistance or enable the reduction of concrete wythes' thickness while maintaining equal fire resistance. The potential use of sandwich walls as fire walls is of great significance for the construction industry.

There is a gap in the normative regulations regarding the fire behaviour and fire resistance of sandwich walls. Furthermore, a classification concept for sandwich walls is lacking, hindering the wide use of PCSPs as fire walls. This paper addresses this knowledge

gap by demonstrating the effect of fire loading on the fire resistance of PCSPs on three full-scale sandwich walls. Importantly, the data from these tests could form the experimental background against which to develop a validated classification concept and criteria for the estimation of the fire resistance of sandwich walls in the future.

The insufficient experimental work performed on sandwich panels with energy-efficient connecting systems, such as GFRP connectors, and the difficulty in directly using standard testing methods, constitute barriers to the fire classification of sandwich panels. Addressing these issues, this experimental work involved the analysis of:

- The potential use of a sandwich wall as a fire wall. The fire resistance of a sandwich wall and the fire behaviour of its components were tested and analysed, taking into account the REI-M [18] requirements for load-bearing fire walls.
- The structural integrity of a sandwich wall under fire and impact load. The mechanical coupling of a facing wythe, unsupported at the base, to a load-bearing wythe must not deteriorate to the extent that it falls off during fire exposure time or impact loading.
- The fire behaviour of GFRP connectors. The mechanical performance of connectors, e.g., their support for the facing concrete wythe and their contribution to the shear and flexural stiffness of the wall and anchorage safety in the presence of fire was investigated. Furthermore, their behaviour in terms of combustibility, fire spread and smoke evolution was analysed. Heat-damaged connectors can leave empty holes in wythes or enhance heat-induced concrete spalling, degrading the space-enclosing function of walls. The impact of these effects on the fire resistance of an entire wall was assessed.
- The composite action of a sandwich wall in the presence of fire. The degree of mechanical coupling of both concrete wythes over time in a fire situation should be analysed. The higher degree of composite action sustained through longer fire exposure times is a substantial contribution to the structural robustness of walls [19] and enhances their fire resistance.

This paper presents an experimental investigation conducted on energy-efficient precast concrete sandwich panels with GFRP connectors exposed to fire under vertical eccentric and impact loading. The primary aim of the tests was the confirmation of the sandwich panels' usability as fire walls by testing them according to standardised fire-wall tests. Furthermore, the fire behaviour of the walls' components and resulting composite action in fire situations was assessed.

## 2. Experimental Program

The experimental investigation described in this paper was aimed at assessment of the fire resistance of steel-reinforced concrete sandwich walls with highly thermally insulating materials and GFRP connectors and fire behaviour of their components. Furthermore, their usability as fire walls and their composite action under fire load are analysed and discussed.

### 2.1. Specimens

The adopted dimensions (3000 mm by 3000 mm—width by height) of the full-scale sandwich walls followed the recommendations of DIN EN 1365-1 [20]. The tested sandwich walls were constructed out of three layers: a load-bearing and facing wythes of steel-reinforced concrete, and a core layer of mineral wool thermal insulating plates and GFRP connectors (see Figure 1). The test specimens were stored more than 100 days after concrete placement and were protected against rainfall. This complied with the requirements of DIN EN 1363-1 [21], according to which test specimens should be stored for long enough for its strength and moisture content to be comparable to those of a member in the usual application conditions. The results of the concrete compressive strength tests of the used concrete are listed in Table 1. The concrete wythes were reinforced with the standard steel grid with cross-sectional area in both directions equal to 1.88 cm$^2$/m, placed in the middle of their cross-sectional depth.

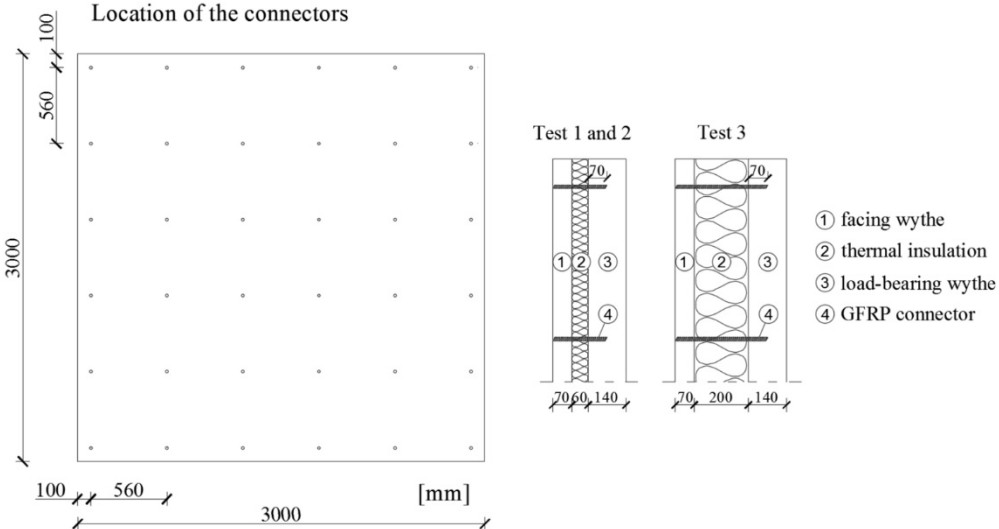

**Figure 1.** Test specimens; location of the connectors in the walls.

**Table 1.** Concrete compressive strength tests.

| No. | | Compressive Strength | Mean Value | Standard Deviation |
|---|---|---|---|---|
| - | (Days) | (N/mm$^2$) | (N/mm$^2$) | (N/mm$^2$) |
| 1. | | 57.92 | | |
| 2. | | 55.27 | | |
| 3. | 182 | 56.59 | 55.78 | ±1.95 |
| 4. | | 53.34 | | |

The thickness of the insulation layer was equal to 60 mm in Tests 1 and 2, and to 200 mm in Test 3. Compression-resistant mineral wool in the form of plates manufactured with actual thickness of the insulation layer (60 or 200 mm) was applied [22]. In the production process, the insulation was placed directly onto the fresh concrete mixture and the top wythe was concreted directly on the insulation. The concrete was then compacted so that the bond of insulation to concrete was comparable to typical three-layered sandwich panels.

The mechanical connection and transfer of shear force between both concrete layers were ensured by bar-type connectors made out of glass-fibre-reinforced polymer [23]. The GFRP connectors were made from electrical-grade corrosion-resistant (E-CR) [24] glass fibre and vinyl-ester resin, with a diameter of approximately 20 μm [25]. The E-CR glass fibre was an alumino-calcium-silicate-based glass fibre with no more than 1% alkali as its acid resistance [26,27]. The location, number and anchorage length of the connectors were the same for all three test specimens. Bars, with a diameter of Ø 12 mm, had ribbed surfaces (which were indented after curing of the resin). Their declared modulus of elasticity was 60 GPa and the transition temperature of the used vinylester resin was specified as 180 °C by the manufacturer. The connectors were cut at the ends at an angle of 30° and their mechanical connection to concrete was ensured by embedding them in concrete during casting (see Figure 2). The clear free length of connectors in Specimens 1 and 2 was equal to 60 mm, while, in Specimen 3, it was equal to 200 mm.

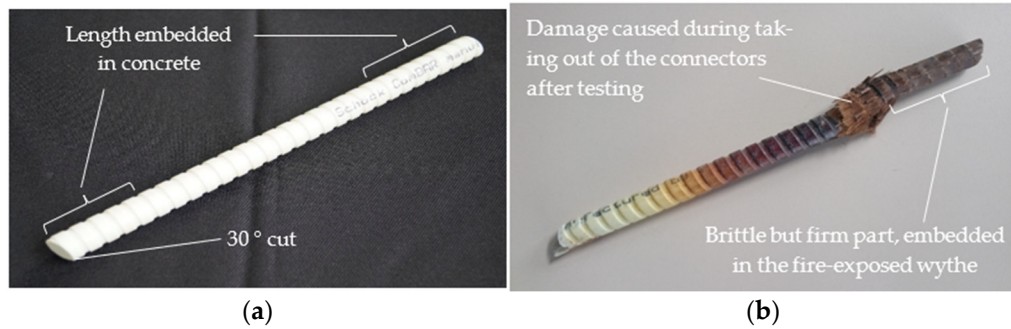

(**a**)                                               (**b**)

**Figure 2.** GFRP connectors used as a shear connection between the two concrete wythes, before assembly in the wall (**a**) and after the fire test—Test 3 (**b**).

### 2.2. Specimen Configuration

The test programme was developed to focus on the fire resistance of sandwich wall and its performance in terms of potential use as a fire wall. Three fire tests were conducted, where the side of the panel being exposed to fire and the thickness of the core layer were the investigated parameters (see Figure 3).

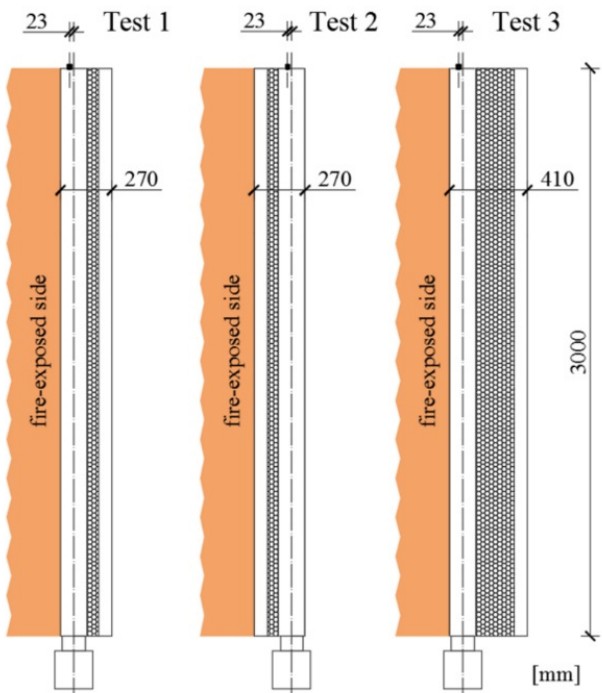

**Figure 3.** Vertical test set-up and location of the eccentric vertical loading.

Two representative limiting cases were tested: thin and thick insulation layer, which were 60 and 200 mm, respectively. Different insulation thicknesses were deemed to cause different temperature increases in the connectors over their lengths and, therefore, different mechanical performances. Additionally, since the concrete wythes were of a different thickness, the side of the wall being exposed to fire could potentially impact the behaviour of connectors. In order to obtain more complete information on the fire resistance of sandwich wall, the two specimens were tested in two configurations, differing in terms of the side of the wall being exposed to fire. This was performed for a wall with 60 mm insulation layer, because in this way, the more severe case could be tested. Specimens with insulation layer of 200 mm were tested only with load-bearing wythe being exposed to fire.

The thickness of the insulation layer in contemporary PCSPs used in many countries varies between 50 and 75 mm, while in France and Germany, panels with thicknesses up

to 200 mm are produced [10]. On the one hand, thicker insulation layers lead to better thermal properties in walls in normal situations. On the other hand, assuming that the cross-sectional geometry and properties remain constant, an increase in length results in a decrease in shear performance. The lower shear stiffness of connectors with longer free lengths results in a decrease in the degree of composite action generated in walls.

Therefore, Specimens 1 and 3, differing in the thickness of their insulation layers, were designed to demonstrate this parameter's influence on wall flexural stiffness in the presence of fire. Thicker thermal insulation, as in the case of Test 3, can cause a longer part of the connector to reach high temperatures. The deteriorated mechanical properties in longer connectors can lead to more severe damage to the shear connection of both concrete wythes than would be the case with short connectors. The heat-induced decrease in connector stiffness and temperatures above the melting point of the resin could result in highly reduced or entirely lost composite action between the concrete wythes under fire conditions. Furthermore, this could lead to impairment of wall integrity and the facing layer falling off the load-bearing layer. This condition, being inacceptable for a fire wall, should be investigated.

The geometries of Specimens 1 and 2 were identical in order to determine full fire resistance of the sandwich wall exposed to fire on each side. The facing wythe is usually of a smaller thickness than the load-bearing wythe in order to reduce its weight. As a result, its insulating properties are much lower. Therefore, in sandwich walls with discrete connectors, where the DCA is relatively low, the structural contribution of the facing wythe to the flexural stiffness is typically neglected [10]. Nevertheless, the structural contribution to both the stiffness and the DCA present can potentially be reduced much more quickly when the facing wythe is exposed to fire. Excessive increases in temperature in connectors can lead to potentially dangerous fall-off of the facing wythe. In this case, any degradation of concrete by fire- or heat-induced spalling of concrete also leads to significant losses in connectors' anchorage strength.

It should be mentioned that the conditions in the test configuration in which the facing layer was exposed to a fully developed fire are less likely to occur. However, this is possible in the case of a building extension when an already existing part of the building is expected to change its function and an external wall becomes an internal one. In order to test all three specimens under equally severe conditions, a full fire curve was applied for fire loading.

### 2.3. Test Setup and Procedure

In order to address the described issues, sandwich walls were tested in a procedure for load-bearing fire walls following the requirements of the testing standards DIN EN 1363-1 [21] and DIN EN 1365-1 [20] (see Figure 4). The fire loading was consistent with the uniform-temperature time-curve (UTTC), according to ISO 834-1:1999 [28]. Observations were made during the tests accounting for criteria specified in the code as well as for potential smoke development and general structural and thermal behaviour. Deformations of the walls, together with temperature at measurement points on the wall, were constantly recorded. The furnace internal temperature was monitored and regulated with six standard-plate thermocouples placed approximately 10 cm away from the fire-exposed surface of the specimen. Adopted loading included 120 min of exposure to fire, vertical eccentric load and three impact loadings applied after turning off the burners. Complementary observations were made after approximately two days of cooling down, when all heat-induced damages were analysed.

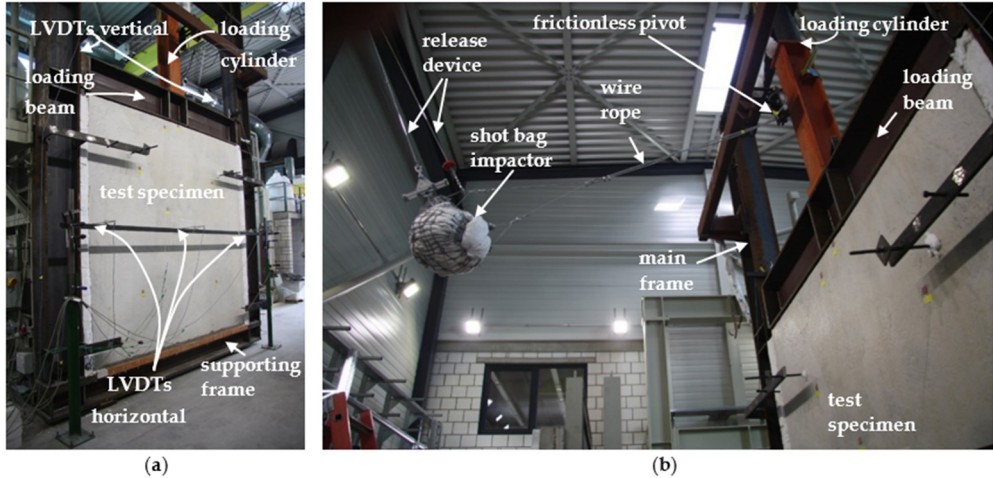

**Figure 4.** (**a**) Experimental set-up for fire tests under vertical eccentric loading. (**b**) Test set-up for the impact load in starting position—as in Test 2.

In each test, the wall was supported against the load-bearing wythe, which was placed directly on the testing frame on a 10–20-millimetre layer of mortar. Following the common and adopted support scheme, facing wythe remained suspended, held up only by the connectors and its bond to the thermal insulation. In this case, failure of connectors can lead to a catastrophic failure, involving fall-off of the facing wythe. This common industrial practice represents more severe testing conditions than those experienced by both fully supported concrete wythes at the base of the wall. In order to evenly distribute the vertical load, the full-cross-section $15 \times 15$ mm$^2$ steel bar was mortared onto the top edge of the load-bearing wythe in the eccentric position (see Figure 3). The vertical gaps between the specimen and the testing frame were sealed with elastic, non-combustible stone wool; therefore, relocatable sealing was used.

### 2.4. Loading

The test load applied during exposure to fire was related to a load resulting from the weight of another upper storey, which would be applied on the wall in the final state. Thus, a line load of 52 kN/m was adopted. This load was applied as a linear, uniformly distributed vertical eccentric load onto the load-bearing wythe with an offset of one sixth of its thickness (140/6~23 mm) in the direction of the furnace interior (see Figure 4a). The vertical load was kept constant until failure with an accuracy of $\pm 5$ kN, which was a result of the load-application system.

After being exposed to fire, the wall was loaded with impact load according to DIN EN 1363-2 [29], which is required for classification as fire walls should be able to withstand the impact of falling deck girders in fire conditions. The special test set-up for impact loading with energy of 3000 Nm is shown in Figure 4b. According to the testing standard, the first two strikes were applied, while the vertical load was still applied with hydraulic cylinder. For the third strike, the wall was unloaded. Relatively stiff connectors in combination with elastic insulation material can cause the facing layer to fail under impact load. For this reason, the location of connectors was designed such that the impact loading hit the middle of the area between connectors' rows to represent worst-case scenario for facing wythe when hit in the mid-span between connectors, which then act as supports.

### 2.5. Measurement

The temperature on the side not exposed to fire was measured with 9 type-K thermocouples (TC1-9) with precision of $\pm 2.5$ °C. The sketch showing the location of thermocouples following the requirements of DIN EN 1363-1 [21] can be found in Figure 5. The thermocouples TC1–5 were used to determine the mean temperature increase, while the thermocouples TC6–9 served to determine the maximal temperature increase on the side

not exposed to fire. While thermocouples TC6–7 followed the recommendations of the testing code, the thermocouples TC8 and TC9 were placed at the positions of connectors in the uppermost (TC8) and the middle (TC9) connector rows to measure the temperature increase considering the potential formation of thermal bridges due to presence of the connectors. Additionally, the temperature at the depth of interface between insulation and wythe not exposed to fire was measured with a thermocouple set in concrete, TC 10 (see Figure 3).

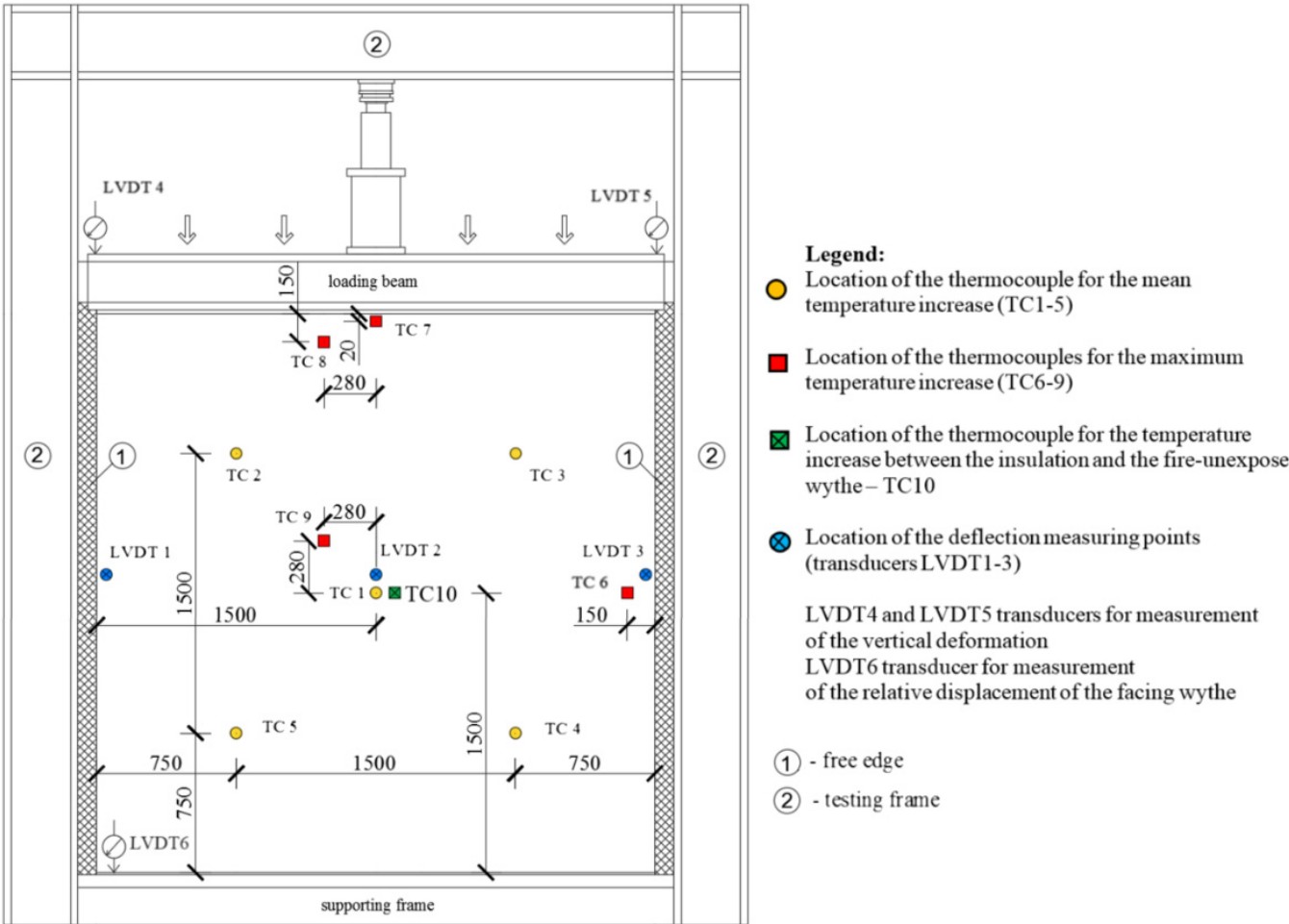

**Figure 5.** Locations of the temperature and deformation measuring points, as well positioning of the test specimen in the testing frame of the furnace, in mm.

The deformations of the walls were measured with linear-variable inductive transducers (LVDTs). The horizontal deformations were measured at the three points at the mid-height of the wall (LVDT 2) in the middle and on both left (LVDT 1) and right (LVDT 3) sides, close to the edges. The locations of the measuring points are shown in Figure 5. The vertical deformations were measured on both left (LVDT 4) and right (LVDT 5) sides of the head of the wall (see Figure 5). Additionally, in Test 1 and Test 3, the relative displacement of the facing wythe in relation to the load-bearing wythe was determined by measuring the vertical displacement of the facing wythe on the bottom edge (LVDT 6). In Test 2, this measurement was not possible because the facing wythe was enclosed within the furnace. The 400-kilonewton load cell was used to control applied vertical load. All data records were taken at the frequency of 0.2 Hz.

### 3. Test Results and Discussion

The tested sandwich walls met the REI requirements over 120 min of fire-exposure time and withstood the three subsequent impact loadings (criterion M) without experiencing any significant damage, considering the analysed criteria. The fire behaviour of the walls was assessed based on the observations made during their exposure to fire and approximately two days after the tests on the cooled-down specimens. The structural and thermal performance of the walls in the fire tests were analysed regarding the standardised criteria set for walls. The results of the structural and heat transfer behaviour are summarised in Table 2. Observations regarding the degree of composite action provided by the shear GFRP connectors were made and are discussed here.

**Table 2.** The performance criteria according to DIN EN 13501-2 for load-bearing members with a space-enclosing function.

|  | Test 1 | Test 2 | Test 3 |
|---|---|---|---|
| **Load-bearing capacity** | | | |
| Axial height reduction (mm) | $9.90 < C^{(1)} = -30$ | $2.05 < C = -30$ | $18.76 < C = -30$ |
| Rate of the axial height reduction (mm/min) | $0.33 < dC/dt^{(2)} = 9$ | $0.27 < dC/dt = 9$ | $0.61 < dC/dt = 9$ |
| Deflection (mm) | $45.38 < D^{(3)} = 160,7$ | $10.23 < D = 160.7$ | $56.89 < D = 160.7$ |
| Deflection rate (mm/min) | $1.40 < dD/dt^{(4)} = 7.14$ | $1.25 < dD/dt = 7.14$ | $2.63 < dD/dt = 7.14$ |
| **Thermal insulation** | | | |
| Mean temperature increase on the side not exposed to fire | $52.7\,°C < 140\,°C$ | $29.7\,°C < 140\,°C$ | $51.0\,°C < 140\,°C$ |
| Maximal temperature increases above the initial temperature | $56.4\,°C < 180\,°C$ | $31.1\,°C < 180\,°C$ | $56.1\,°C < 180\,°C$ |

[1] is the maximal allowable axial height reduction in mm, [2] is the maximal allowable rate of the axial reduction in mm/min, [3] is the maximal allowable deflection in (mm), [4] is the maximal allowable deflection rate in mm/min.

#### 3.1. Structural Behaviour

The results of the out-of-plane and axial deformation of the tested walls are summarised in Table 2 and compared against the limit values given in [18]. The fire resistance of the load-bearing fire wall was deemed to have failed if the limit deformations in the fire test were exceeded. The horizontal deflection, axial deformation, and the rates of change of these factors in each wall were far below the allowable values.

#### 3.2. Axial Displacement Behaviour

Before the furnace ignition, a small initial reduction in the wall height under applied vertical load was observed in Tests 2 and 3 (LVDT 4 and 5, see Figure 6). The small initial expansion of the wall in the case of Test 1 was caused by the short pre-heating of the furnace, caused by technical problems with the furnace controller. However, soon after the ignition of the furnace burners, the wall experienced heat-induced expansion, which induced pressing against the load introduction beam placed between the wall head and the hydraulic cylinder with the load cell. To reduce the unnecessary increase in the applied load, the oil pressure was adjusted during the test to keep the load constant. The recorded temperature-induced expansion was at its most intensive in the case of Specimen 3, which confirms that the thicker insulation caused the fire-exposed load-bearing layer to be more thermally loaded than in the case of Specimen 1. Entirely different behaviour was observed in the case of Specimen 2, where the insulation and fire-exposed facing wythe acted as a fire protection for the vertically loaded load-bearing wythe, which experienced expansion of around 1 mm, which is equal to only about 10% and 7% of the thermal axial expansion of Specimens 1 and 3 at the end of the test, respectively.

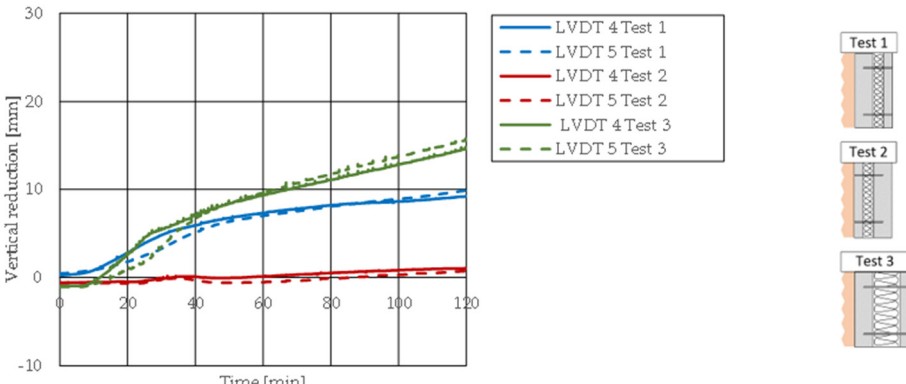

**Figure 6.** Deformation recordings in the vertical direction measured at the wall's head.

It should be mentioned that the axial deformation observed in the tests was opposite (extension) to the one foreseen in the testing standards (the limit values refer to the reduction in the height) [30]. Nevertheless, the final value and rate of these deformations did not exceed the allowable values.

*3.3. Out-of-Plane Behaviour*

In each test, a clear, heat-induced bowing of the wall inwards, in the direction of the furnace, was observed. The recordings of the horizontal deflection are shown in Figure 7. The monitoring of the horizontal deflection was prescribed by the testing code, but it was also important also because relatively large deformations were expected. The one-sided fire exposure of the concrete sandwich walls resulted in heat-induced bowing, which was expected to be larger than in the case of solid concrete walls of the same thickness due to the larger temperature gradient between the fire-exposed and non-fire-exposed sides.

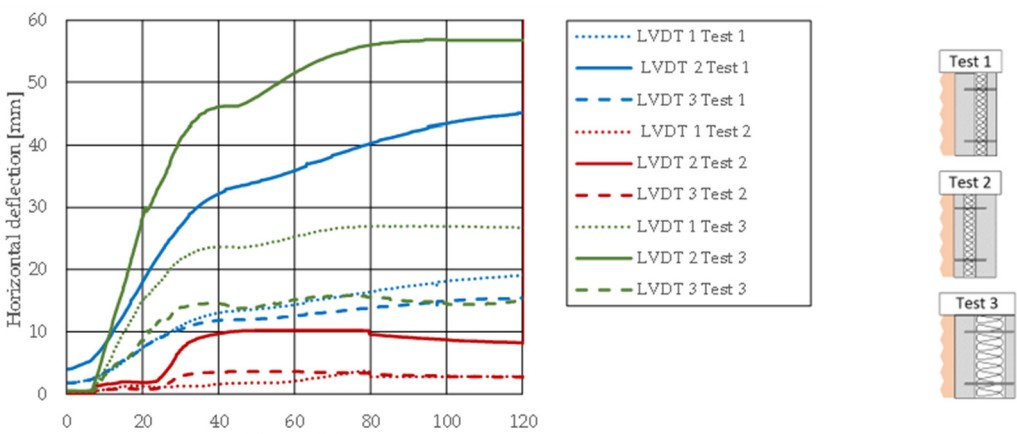

**Figure 7.** Horizontal deformations.

A phase of intensive deformation was observed on each test 7–35 min after the furnace ignition. Subsequently, the rate of horizontal deformation tended to decrease. In each test, the temperature between the insulation and the wythe not exposed to fire (see Figure 8, TC10) did not rise above the level of 100 °C; this was primarily due to the boiling water in the concrete and mineral wool. The observed deflection resulted from the distributions of the strains between the unheated load-bearing wythe and the facing wythe, which underwent large heat-induced expansion. The bowing of the walls was also caused by the bowing of particular wythes resulting from the temperature difference between their sides. The small (2–4 mm) initial deflection in the case of Test 1 was caused by short pre-heating of the furnace, caused by technical problems with the furnace controller.

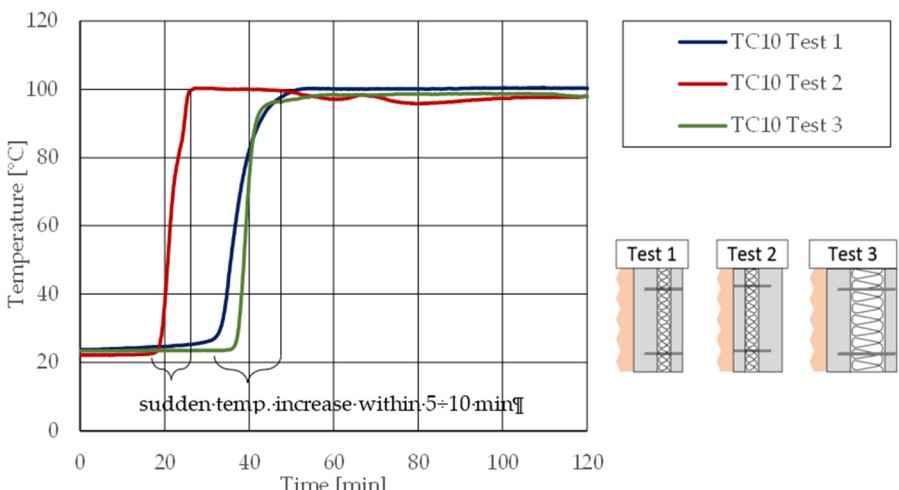

**Figure 8.** Temperature in the plane between the insulation and the concrete wythe not exposed to fire.

The largest deflection was observed in the case of Specimen 3 (56.89 mm). This can be explained by two effects. Firstly, the degree of composite action and the flexural stiffness in this case were deemed to be lower as the connectors with free lengths of 200 mm had lower shear stiffness than those in Specimens 1 and 2. Secondly, the thicker insulation changed the temperature gradient in the fire-exposed wythe, resulting in higher temperatures throughout its cross-section and a higher temperature difference between the two wythes, which in turn resulted in larger thermal expansion and bowing. The temperature gradient through the fire-exposed wythe, however, was not measured.

Following this approach, the smallest deflection, as expected, was observed for Specimen 2 (10.23 mm). The lower thickness of the fire-exposed wythe resulted in a lower temperature gradient and less thermal bowing than in Tests 1 and 3. The relatively low stiffness of the facing wythe, as well as its partial disconnection from the load-bearing wythe (due to the failure of the connectors in the inner area of the wall) and the concrete spalling observed in this test, resulted in a much smaller deflection of the wall in Test 2 until a time of 25 min. Subsequently, once the gradually heated load-bearing wythe started to thermally bow itself outward, the deflection gradually proceeded (compare Figure 9).

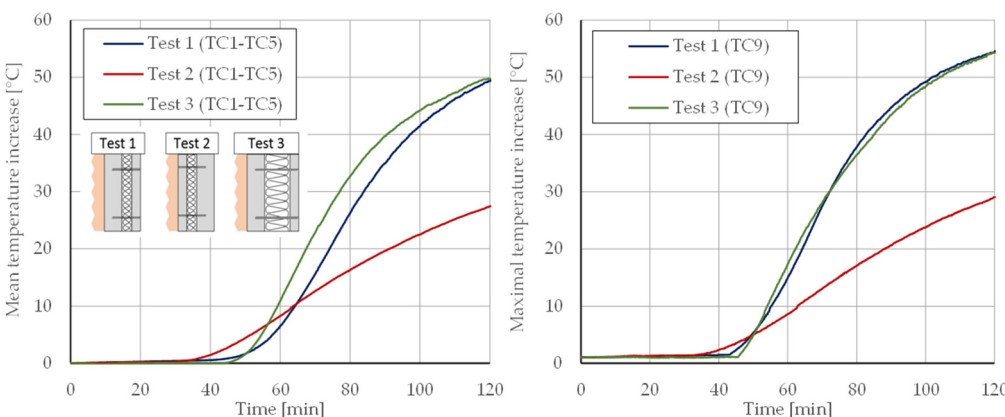

**Figure 9.** The mean temperature increases on the side not exposed to fire.

### 3.4. Heat-Transfer Behaviour

The heat transfer from the fire-exposed side to the fire-unexposed side was monitored with thermocouples TC1–10. Considering the requirements of structural members, which should serve as fire walls, the mean temperature on the fire-exposed side must not exceed 140 °C, while the maximum temperature on the side not exposed to fire must not exceed 180 °C above the initial temperature. Although testing standards do not set requirements

for the temperature increase within the cross-sections of walls, knowledge about the temperature gradient through the wall to the side not exposed to fire enables a better understanding of the thermal and mechanical performances of insulation and connectors.

A sudden increase in temperature to around 100 °C at the depth of the insulation and non-fire-exposed concrete wythe interface was observed (see Figure 8). This indicates that after the temperature of 100 °C was reached in the fire-exposed wythe, the moisture escaping the concrete and insulation increased the temperature throughout the thickness of the insulation in a very short time of ca. 5–10 min. The observed plateau at around 100 °C can be explained by the evaporation of free water at the interfacial depth, as a similar effect was observed in the fire test on sandwich panels in [15]. This phenomenon should be taken into consideration, especially in the case of temperature-sensitive insulating materials, where sudden increases in temperature up to 100 °C can entirely eliminate the insulating layer from the structural contribution to the flexural stiffness of sandwich panels. The faster increases in temperature in the case of Specimen 2 (after 18 min) were caused by the thinner concrete wythes protecting the insulation from heat, indicating that the thickness of fire-exposed concrete wythes can have a crucial influence on the fire resistance of heat-sensitive insulation materials in sandwich walls. The small decreases in the temperature of the interface, from 100 °C to 96 °C, recorded in Test 2 may have been caused by the evaporation of water from the insulation layer and the removal of the main heat-transferring agent at this stage of the fire.

The mean temperature increase measured as a mean value of the temperatures recorded with thermocouples TC1 to TC5 in each test is presented in Figure 9 (TC1–5). Specimens 1 and 3, which differed only in their insulation thickness, showed similar heat transfer behaviour. In both tests, the intensive temperature increase phase started approximately 50 min after the furnace ignition and, therefore, at the end of the tests, almost the same temperature (difference of ~1 °C) was reached. However, the maximum mean temperature in the case of Specimen 3 reached only 51.0 °C (only 37% of the maximum allowable mean temperature). The mean temperature increased more quickly in the case of Specimen 3, which had a thicker insulation layer. This indicates that the thickness of the thermal insulation does not play a leading role in terms of the heat transfer under fired conditions when boiling water needs to escape the wall. The relatively small difference in heating rate between Test 1 and 3 could have been caused by a difference in moisture content in the concrete and the insulation. The moisture content was not measured; however, a few litres of condensed water steam leaked out of Specimens 1 and 3 outside the furnace, and intense evaporation was visible on the perimeters of the walls.

In the case of Sandwich 2, where both the facing wythe and the insulation were located directly in the furnace, the temperature increase started 5 min sooner, at 40 min after the furnace ignition, and reached 29.7 °C, only 21% of the maximal allowable mean temperature. This can be explained by the escape of boiling moisture to the inside of the furnace, leading to the relatively fast drying of the insulation and slower heat transfer to the side of the wall not exposed to fire. Secondly, the mean temperature increased on the unexposed side relatively slowly as the unexposed load-bearing wythe was heated mainly solely with water boiling at 100 °C. Although the thermal insulation fulfilled its function in a fire situation, the early heating of the entire insulation layer can occur, as observed. This effect can be delayed by increasing the concrete wythe thickness, preventing the insulation from warming up.

The maximum temperature for each specimen was recorded with thermocouple TC9 (see Figure 9), which was located close to the middle of the wall at the connector. This observation confirms that the connectors caused a thermal bridging effect and contributed to the heat transfer to the side not exposed to fire. However, the magnitude of this effect was relatively small. The maximal temperatures were only 56.4 °C and 56.1 °C for Specimen 1 and 3, respectively, and 31.1 °C for Specimen 2, i.e., ~31% (Test 1 and 3) and ~17% (Test 2) of the maximal allowable temperature increase, respectively.

All three tested walls withstood the fire tests and subsequent impact loading. The load-bearing capacity, the space-enclosing and thermally insulating functions were maintained in compliance with the criteria listed in DIN EN 13501-2 [18] for more than 120 min.

*3.5. Connectors and Insulation*

In a fire situation, the insulation layer should first ensure the thermal protection of the load-bearing layer, stay firm during the fire, and not contribute to fire spread. Further, specifically for sandwich walls, the effectiveness of shear transfer during fire provided by the insulation and connectors should be assessed, since this gives information about their contribution to the composite action. This was achieved based on the observations made on the wall's deformations and its general physical behaviour during and after its exposure to fire.

No debonding of the mineral wool from the concrete on the visible outer edges of the walls in Tests 1 and 3 was observed. The deformation of the insulation followed the curvature of the bowed concrete wythes without creating any visible diagonal shear cracks. To the naked eye, upon physical examination, the insulation remained firm during and after its exposure to fire. However, the observation made on the cooled specimens after firing showed that the mineral wool crumbled and became brittle on the fire-exposed side to a depth of up to half of its thickness, especially in the middle of the wall, where its exposure to high temperatures was at its most intense. In Test 2, the fire-exposed facing wythe lost its mechanical connection to the load-bearing wythe in the vast middle area of the wall and, therefore, its bond to the thermal insulation. In this case, however, the insulation remained bonded to the load-bearing wythe and its thermal behaviour was similar to that in the two other tests (Figure 9).

The post-fire condition of the connectors was investigated by cutting the specimens in half along their height (see Figure 10). Observations made with the naked eye showed that the connectors in the uppermost and the lowermost row (position A) remained in their original colour and shape, without revealing any visible signs of heat-induced damage. By contrast, the connectors of the inner rows (position B) experienced visible damage due to their exposure to high temperatures. The magnitude of this deterioration varied across the tests.

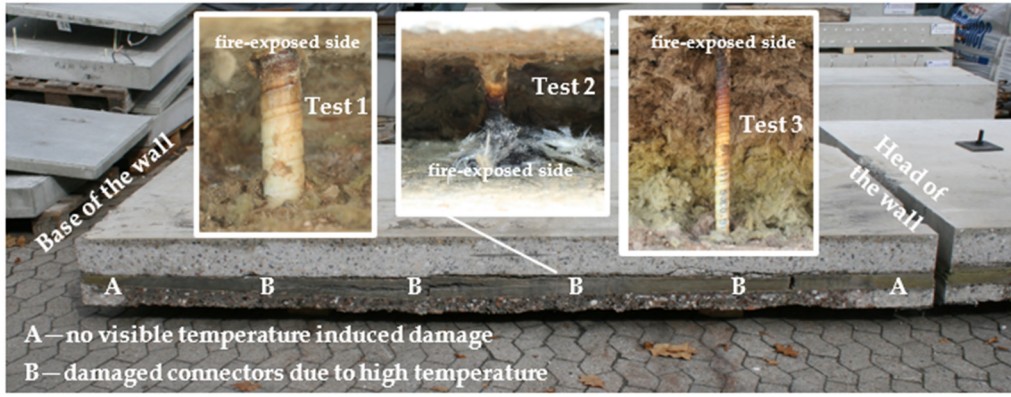

**Figure 10.** Location of the melted connectors in the cut sandwich wall after fire tests—example of Specimen 2 and typical damage state of connectors.

The connectors in Specimens 1 and 3 in position B were substantially damaged along their free length, close to the fire-exposed wythe. In these parts, visible partial carbonation and the absence of resin on the perimeter of the connectors was observed, leaving some fibres unprotected. Similar, but much more severe damage was observed in Specimen 3, where the connectors were partially carbonated on the longer part of their free lengths and the damage to the resin protection close to the fire-exposed wythe was larger. In both cases, while the connectors stayed firm and in their original shape, the glass fibres in their cores must have still been protected by the unmelted resin. The length of the described

visible heat-induced deterioration was equal to ca. 20 mm and 70 mm for Specimens 1 and 3, respectively. This difference could be explained by the greater increase in temperature in the longer parts of the connectors in Specimen 3 due to their significantly thicker thermal insulation (200 mm).

In order to better identify the damage, the connectors were taken out of the concrete from the middles of Specimens 1 and 3, where they appeared to still be firm and rigid upon hand inspection, even along their length, and embedded in the fire-exposed concrete wythe. The partial fracture and exposure of the glass fibres visible in Figure 2 occurred unintentionally during the removal process, which may indicate that the connectors' mechanical properties significantly deteriorated. This observation goes far beyond the results reported in [19], where the GFRP rebars made from the same material as the tested connectors did not show any visible degradation of their external surfaces after reaching a temperature above the glass transition temperature $T_g$ of the resin. However, a strong reduction in bond strength, whose main governing parameter is the temperature near the resin $T_g$ [31], has a stronger influence than surface finishing and diameter, as reported in [32]. Although, at the end of fire exposure, the temperatures in the exposed wythe must have been much higher than the $T_g$ of the connectors' resin, no signs of pull-out and no premature collapse of the facing wythe due to the debonding of the GFRP connectors were observed. This finding agrees with the results obtained in [31,33], as the bonds of the ribbed bars from the same manufacturer were less affected than the sand-coated GFRP bars reported in [34].

In Test 2, the exposure of the facing wythe (in this case, the 70 mm thickness was further reduced due to concrete spalling) to fire resulted in much more severe damage of the connectors in position B. In this case, the resins of the connectors melted up to ca. 50 mm of their free length and over the length to which they were embedded in the fire-exposed wythe. This was noted during the examination of the cooled specimen, when the resin in this region was not present and only the unprotected continuous intact glass fibres were visible. Although the connectors experienced some heat-induced damage over their entire free length, substantial carbonation was observed up to about half of their free length. Therefore, it could be concluded that the melted connectors in all the inner rows lost their entire load-bearing capacity and the facing wythe was supported solely by the upper and bottom rows of the unmelted connectors (position A).

Sudden and high increases in temperature in the concrete layer led to excessive pore pressure and compressive stress on the fire-exposed side, leading to explosive spalling. In the tested fire-exposed wythes, the temperature could have been larger than would have been the case for single wythes of the same thickness, as a result of the thermal insulation preventing the heat transfer outside the furnace. The spalling of the fire-exposed layer of concrete occurred in Tests 2 and 3 up to depths of ~40 mm within 15 to 20 min of the furnace ignition. The lack of spalling in Specimen 1 could have been due to differences in the moisture content of the concrete, which was itself probably caused by different conditions during the curing and drying times, which were not recorded. The influence of the connectors on the spalling behaviour of the concrete wythes was analysed. No spalling-initiating effect caused by the presence of the connectors could be identified. An examination of the post-fire surface revealed that in Test 2, the areas surrounding the locations of the connectors showed smaller or no spalling comparing to other areas on the same specimen. This could have been caused by lower pore pressure around the empty holes left by the melted connectors' sections, where moisture could escape the concrete.

### 3.6. Composite Action

The mechanical connection of concrete wythes provided by insulation and connectors is crucial for the fire resistance of sandwich walls for two reasons: First, the composite action provided by the shear transfer between the concrete wythes increases the flexural stiffness of the wall, and second, burning debris or pieces of the facing wythe must not fall off the load-bearing wythe of the sandwich wall applied as a fire wall. The effectiveness of the connection of the two concrete wythes was monitored with LVDT6 in Test 1 and 3,

which measured their relative displacement. A general assessment of the composite action was performed on the basis of these data and of observations of the wall's deformations during exposure to fire.

The failure of the connectors should result in a displacement downwards, since the shear stiffness of the core layer is weakened by exposure to fire. On the other hand, the thermal bowing of the entire wall and the rotation of the cross-section at the wall base should result in the upward displacement of the bottom edge. The displacement recorded with LVDT6, which was the result of the two described effects, showed that the facing wythe's bottom edge was moving upwards (see Figure 11). This indicates that the displacement of the edge resulting from the rotation caused by the heat-induced bowing was clearly of a larger magnitude than the potential sagging of the facing wythe under its weight supported by the core layer.

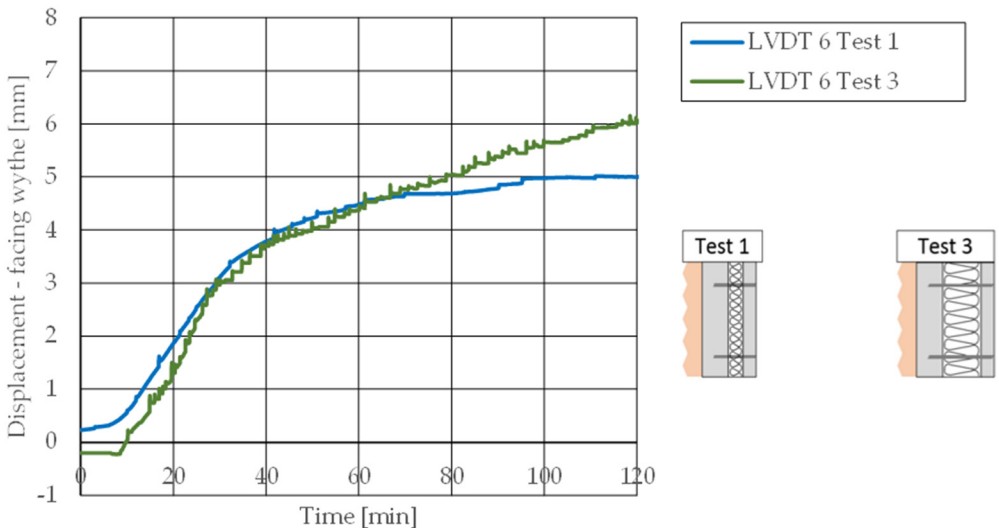

**Figure 11.** Relative displacement of the facing wythe.

As schematically depicted in Figure 12, the thermal elongation of the fire-exposed side of the wall induced bowing in the direction of the furnace. The edge of the facing wythe following the curvature of the deforming load-bearing wythe was moving upwards, since the rotation point was at the bottom edge of the load-bearing wythe. The small initial downward displacement (−0.2 mm) in the case of Test 3, visible in Figure 11, was caused by the applied vertical load. This phase was skipped in the case of the Test 1 (where the initial displacement was 0.3 mm) due to problems with the furnace controller resulting in a minor initial pre-heating of the wall. In both tests, the displacement of the facing wythe began an intensive growth phase 9 min after the furnace ignition, when the walls underwent heat-induced bowing. The final relative displacement of the facing wythe's edge was larger by 1 mm in Test 3 due to the thicker cross-section of the wall and, therefore, larger temperature differences between the concrete wythes, leading to larger horizontal deflection (see Figure 7) and larger rotation at the bottom edge of the wall.

As indicated in [15] the facing wythe is assumed to provide thermal and non-structural protection to the load-bearing layer in the presence of fire. This function was fulfilled throughout the duration of the tests, even on Test 2. Despite the local damage to the connectors, the facing wythe and the insulation stayed in place. The observed behaviour of Specimens 1 and 3 indicates that the degree of composite action was sufficient to follow the curvature of the deforming wall by both concrete wythes and keep them in a steady relative position. In Tests 1 and 3, in the visible cross-section of the facing wythes sticking outside of the furnace, a few regularly distributed curvature-induced cracks with spacing of approximately 500 mm were observed. The cracking of the unexposed wythe was interpreted to mean that the connectors and insulation provided the wythes with a connection

that was sufficient to ensure the equal curvature of the deformed wythes. The mechanical coupling of both wythes was provided, and both facing and insulation layers supported the flexural strengthening of the load-bearing wythe. In this case, the flexural stiffness of the deformed unexposed wythe imposed by the thermal extension of the exposed wythe acted counter to the deflection caused by the fire loading, resulting in the reduced deflection of the entire wall. Therefore, it can be inferred that the tested walls acted as partially composite sandwich walls throughout the fire exposure time and during the impact loading.

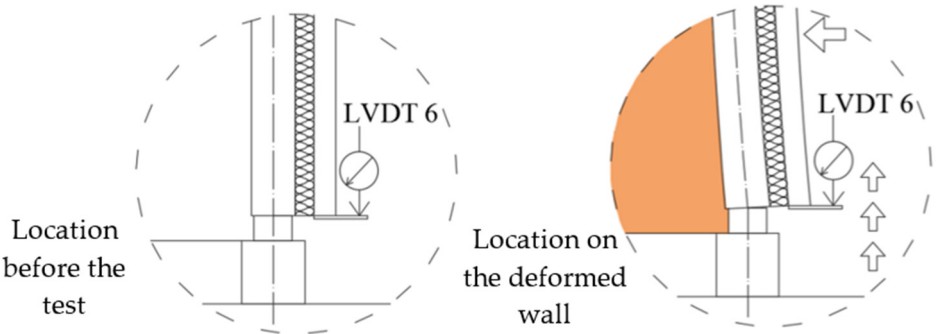

**Figure 12.** Location of the transducers for the measurement of the relative displacement of the facing wythe in the cases of Tests 1 and 3—an example from Test 1.

The quantitative contribution of the insulation to this degree of composite action was not measured. However, since its share can be significant, special care should be paid to the temperature-dependent behaviour of the insulation material used. As reported in [13], the use of other insulation materials, such as expanded polystyrene, can result in the very quick failure of the mechanical function of insulation at about 90 °C in the layer, significantly lowering the flexural stiffness of the panel. Since insulation can rapidly deteriorate after reaching its critical temperature, the mechanical performance of sandwich walls in terms of ultimate load and serviceability load are strongly dependent on the maximal temperature of the fire-exposed side of the panel, not on the heating rate or exposure time. Therefore, the observations made can apply only to walls insulated with fire-resistant mineral-wool insulation.

## 4. Summary and Conclusions

The experimental work presented in this paper was aimed at the investigation of the fire resistance of sandwich walls with GFRP connectors. The structural performance of the three one-storey walls was tested under vertical eccentric loading, exposure to fire and subsequent impact load. The thermal and structural performance of the insulation and connectors under high temperatures were analysed. All the REI requirements were met for 120 min of fire loading and impact loading.

The connectors in the middle area of Specimens 1 and 3 experienced significant heat-induced damage but were still able to mechanically couple both concrete wythes. In Specimen 2, the connectors melted through the entire thickness of the fire-exposed facing wythe. In this case, however, the wall maintained its structural integrity due to the still load-bearing connectors at its perimeter. Due to the potential failure of all the connectors, the application of additional, fire-resistant connectors to support the load-bearing function in the fire situation should be considered.

The possible influence of the connectors on the heat-induced spalling of the concrete was analysed. In Test 2, a spalling-prevention effect caused by the melted connectors was observed. The concrete in some areas surrounding the empty holes left by the connectors on the fire-exposed side did not spall during the test, which could suggest that the moisture was able to escape through the voids left by the connectors, lowering the pore pressure in the concrete.

Therefore, it can be concluded that the presence of GFRP connectors did not contribute to the concrete spalling but, on the contrary, had a mitigating influence on the surrounding concrete. The space-enclosing function of the wall was fully maintained and the spread of fire and smoke development were prevented.

The moisture enclosed in the wall was brought to boiling and served as a main heat carrier through the insulation at the early stage of the test. This resulted in the heating-up of the entire insulation and the internal side of the wythe not exposed to fire to 100 °C at almost the same rate for all the specimens. This sudden increase at an early stage of fire exposure should be considered in the application of different temperature-sensitive or combustible materials, since the insulation can be eliminated from thermal and mechanical contributions in this way.

The recorded displacement of the unsupported facing wythe showed that the wythes followed the same curvature of heat-induced bowing. This indicates that the connectors maintained their mechanical properties throughout the testing time and were able to maintain a constant distance between the concrete wythes, coupling them together. Additionally, no indication of the potentially dangerous fall-off of the facing wythe in Test 2 was observed. However, the curvature-induced cracking of the wythes not exposed to fire was observed in Test 1 and 3.

The structural coupling of the two concrete wythes provided by the shear stiffness of the bar-type connectors and the insulation was observed in all the tests. Even in Test 2, where the connectors in the middle area of the wall were damaged, the facing wythe remained mechanically attached to the load-bearing wythe. This indicates that the mechanical contribution of the core and the facing layers to the flexural stiffness can be significant. The large thermal bowing observed in Tests 1 and 3 suggests that some degree of composite action was maintained throughout the fire testing time and during the mechanical impact loading.

Sandwich walls can maintain their structural and fire-protective functionality under fire conditions not only due to their excellent thermal insulating properties, but also due to their significantly higher flexural stiffness compared with concrete walls with structurally inactivated hanging cladding systems. All the tested sandwich walls, on the basis of the test results, could be classified in fire resistance class REI 120—M.

**Author Contributions:** Conceptualization, M.H., M.P. and C.T.; methodology, M.H., M.P. and C.T.; investigation, M.H.; writing—original draft preparation, M.H.; writing—review and editing, M.H., M.P., C.T. and S.G.; visualization, M.H. and S.G.; supervision, M.P. and C.T.; project administration, M.H., M.P. and C.T.; funding acquisition, M.P. and C.T. All authors have read and agreed to the published version of the manuscript.

**Funding:** This study was funded by Bundesministerium des Innern, für Bau und Heimat (German Government), grant number SWD-10.08.18.7-17.25.

**Data Availability Statement:** Not applicable.

**Acknowledgments:** The authors would like to acknowledge the financial support of the German Government (Bundesministerium des Innern, für Bau und Heimat), project no. SWD-10.08.18.7-17.25, as well as Schöck Bauteile GmbH and Fehr Technologies Deutschland GmbH & Co. KG for providing the materials and the preparation of the specimens.

**Conflicts of Interest:** The authors declare no conflict of interest.

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
