# Peer review of "Experimental Investigation of Concrete Sandwich Walls with Glass-Fiber-Composite Connectors Exposed to Fire and Mechanical Loading"

_applsci, doi:10.3390/app12083872_

Round 1

Reviewer 1 Report

The paper investigated the PCSP with different wall thickness and the connection of glass fiber composite connector when exposed to fire and mechanical loading. The finding is meaningful and interesting. The manuscript is well prepared. It is suggested to be accepted after minor revisions.

(1) For the introduction, it is suggested to add more related refs in the background part.

(2) The experimental setup in Figure 4 is not well descripted. Please point out which the specific name of each part, especially for Figure 4(b).

(3) It is suggested to provide what are the materials of glass fiber composite. Please add what kind of glass fiber it is. Is it silica-based or alumino-based fiber?

(4) Still some typos in the manuscript, such as “Therefore, is could be 495 concluded that”. Please check about it carefully.

(5) The conclusions should be more concise.

Reviewer 2 Report

In this paper, the respected authors have conducted an experimental investigation on the structural and thermal behaviour of precast concrete sandwich panels with mineral wool insulation and glass fiber reinforced polymeric bar connectors. Three full-size walls have been tested under fire, vertical eccentric, and post-fire mechanical impact load in order to assess the connectors’ fire behaviour. This paper is interesting and it is written well. However, a revision is essential before publication.

  • In Fig. 6, the term “vertikal” should be replaced with “vertical”.
  • In Figs. 6 and 7, only displacement-time curves are shown. The authors should add the load-displacement curves. Load-displacement curves are more usual and give better structural information.
  • The conclusion is so lengthy and should be condensed. Give only main finding of the work in this section explicitly.
  • It is not clear that how much does increase the glass fiber composite connectors the fire resistance of concrete sandwich walls (in comparison to traditional connectors)? Please give more details and data in this regard.
  • Literature review on concrete sandwich walls is not complete. The recent researches related to concrete sandwich walls should be added in section 1. Some examples:

 https://doi.org/10.1016/j.compstruct.2018.11.087 , https://doi.org/10.1016/j.engstruct.2021.112149

Round 2

Reviewer 2 Report

The paper is recommended for publication at the present format.